# Geometric Dirichlet Means algorithm for topic inference

**Mikhail Yurochkin**
Department of Statistics
University of Michigan
`moonfolk@umich.edu`

**XuanLong Nguyen**
Department of Statistics
University of Michigan
`xuanlong@umich.edu`

## Abstract

We propose a geometric algorithm for topic learning and inference that is built on the convex geometry of topics arising from the Latent Dirichlet Allocation (LDA) model and its nonparametric extensions. To this end we study the optimization of a geometric loss function, which is a surrogate to the LDA's likelihood. Our method involves a fast optimization based weighted clustering procedure augmented with geometric corrections, which overcomes the computational and statistical inefficiencies encountered by other techniques based on Gibbs sampling and variational inference, while achieving the accuracy comparable to that of a Gibbs sampler. The topic estimates produced by our method are shown to be statistically consistent under some conditions. The algorithm is evaluated with extensive experiments on simulated and real data.

## 1 Introduction

Most learning and inference algorithms in the probabilistic topic modeling literature can be delineated along two major lines: the variational approximation popularized in the seminal paper of Blei et al. (2003), and the sampling based approach studied by Pritchard et al. (2000) and other authors. Both classes of inference algorithms, their virtues notwithstanding, are known to exhibit certain deficiencies, which can be traced back to the need for approximating or sampling from the posterior distributions of the latent variables representing the topic labels. Since these latent variables are not geometrically intrinsic — any permutation of the labels yields the same likelihood — the manipulation of these redundant quantities tend to slow down the computation, and compromise with the learning accuracy.

In this paper we take a convex geometric perspective of the Latent Dirichlet Allocation, which may be obtained by integrating out the latent topic label variables. As a result, topic learning and inference may be formulated as a convex geometric problem: the observed documents correspond to points randomly drawn from a *topic polytope*, a convex set whose vertices represent the topics to be inferred. The original paper of Blei et al. (2003) (see also Hofmann (1999)) contains early hints about a convex geometric viewpoint, which is left unexplored. This viewpoint had laid dormant for quite some time, until studied in depth in the work of Nguyen and co-workers, who investigated posterior contraction behaviors for the LDA both theoretically and practically (Nguyen, 2015; Tang et al., 2014).

Another fruitful perspective on topic modeling can be obtained by partially stripping away the distributional properties of the probabilistic model and turning the estimation problem into a form of matrix factorization (Deerwester et al., 1990; Xu et al., 2003; Anandkumar et al., 2012; Arora et al., 2012). We call this the linear subspace viewpoint. For instance, the Latent Semantic Analysis approach (Deerwester et al., 1990), which can be viewed as a precursor of the LDA model, looks to find a latent subspace via singular-value decomposition, but has no topic structure. Notably, the RecoverKL by Arora et al. (2012) is one of the recent fast algorithms with provable guarantees coming from the linear subspace perspective.

The geometric perspective continues to be the main force driving this work. We develop and analyze a new class of algorithms for topic inference, which exploits both the convex geometry of topic models and the distributional properties they carry. The main contributions in this work are the following: (i) we investigate a geometric loss function to be optimized, which can be viewed as a surrogate to the LDA's likelihood; this leads to a novel estimation and inference algorithm — the Geometric Dirichlet Means algorithm, which builds upon a weighted k-means clustering procedure and is augmented with a geometric correction for obtaining polytope estimates; (ii) we prove that the GDM algorithm is consistent, under conditions on the Dirichlet distribution and the geometry of the topic polytope; (iii) we propose a nonparametric extension of GDM and discuss geometric treatments for some of the LDA extensions; (v) finally we provide a thorough evaluation of our method against a Gibbs sampler, a variational algorithm, and the RecoverKL algorithm. Our method is shown to be comparable to a Gibbs sampler in terms of estimation accuracy, but much more efficient in runtime. It outperforms RecoverKL algorithm in terms of accuracy, in some realistic settings of simulations and in real data.

The paper proceeds as follows. Section 2 provides a brief background of the LDA and its convex geometric formulation. Section 3 carries out the contributions outlined above. Section 4 presents experiments results. We conclude with a discussion in Section 5.

## 2 Background on topic models

In this section we give an overview of the well-known Latent Dirichlet Allocation model for topic modeling (Blei et al., 2003), and the geometry it entails. Let $\alpha \in \mathbb{R}_+^K$ and $\eta \in \mathbb{R}_+^V$ be hyperparameters, where $V$ denotes the number of words in a vocabulary, and $K$ the number of topics. The $K$ topics are represented as distributions on words: $\beta_k|\eta \sim \mathrm{Dir}_V(\eta)$, for $k = 1, \ldots, K$. Each of the $M$ documents can be generated as follows. First, draw the document topic proportions: $\theta_m|\alpha \sim \mathrm{Dir}_K(\alpha)$, for $m = 1, \ldots, M$. Next, for each of the $N_m$ words in document $m$, pick a topic label $z$ and then sample a word $d$ from the chosen topic:

$$z_{n_m}|\theta_m \quad \sim \quad \mathrm{Categorical}(\theta_m); \; d_{n_m}|z_{n_m}, \beta_{1\ldots K} \sim \mathrm{Categorical}(\beta_{z_{n_m}}). \tag{1}$$

Each of the resulting documents is a vector of length $N_m$ with entries $d_{n_m} \in \{1, \ldots, V\}$, where $n_m = 1, \ldots, N_m$. Because these words are exchangeable by the modeling, they are equivalently represented as a vector of word counts $w_m \in \mathbb{N}^V$. In practice, the Dirichlet distributions are often simplified to be symmetric Dirichlet, in which case hyperparameters $\alpha, \eta \in \mathbb{R}_+$ and we will proceed with this setting. Two most common approaches for inference with the LDA are Gibbs sampling (Griffiths & Steyvers, 2004), based on the Multinomial-Dirichlet conjugacy, and mean-field inference (Blei et al., 2003). The former approach produces more accurate estimates but is less computationally efficient than the latter. The inefficiency of both techniques can be traced to the need for sampling or estimating the (redundant) topic labels. These labels are not intrinsic — any permutation of the topic labels yield the same likelihood function.

**Convex geometry of topics.** By integrating out the latent variables that represent the topic labels, we obtain a geometric formulation of the LDA. Indeed, integrating $z$'s out yields that, for $m = 1, \ldots, M$,

$$w_m|\theta_m, \beta_{1\ldots K}, N_m \sim \mathrm{Multinomial}(p_{m1}, \ldots, p_{mV}, N_m),$$

where $p_{mi}$ denotes probability of observing the $i$-th word from the vocabulary in the $m$-th document, and is given by

$$p_{mi} = \sum_{k=1}^K \theta_{mk}\beta_{ki} \text{ for } i = 1, \ldots, V; \; m = 1, \ldots, M. \tag{2}$$

The model's geometry becomes clear. Each topic is represented by a point $\beta_k$ lying in the $V - 1$ dimensional probability simplex $\Delta^{V-1}$. Let $B := \mathrm{Conv}(\beta_1, \ldots, \beta_K)$ be the convex hull of the $K$ topics $\beta_k$, then each document corresponds to a point $p_m := (p_{m1}, \ldots, p_{mV})$ lying inside the polytope $B$. This point of view has been proposed before (Hofmann, 1999), although topic proportions $\theta$ were not given any geometric meaning. The following treatment of $\theta$ lets us relate to the LDA's Dirichlet prior assumption and complete the geometric perspective of the problem. The Dirichlet distribution generates probability vectors $\theta_m$, which can be viewed as the (random) *barycentric coordinates* of the document $m$ with respect to the polytope $B$. Each $p_m = \sum_k \theta_{mk}\beta_k$ is a vector of cartesian coordinates of the $m$-th document's multinomial probabilities. Given $p_m$, document $m$ is

generated by taking $w_m \sim \text{Multinomial}(p_m, N_m)$. In Section 4 we will show how this interpretation of topic proportions can be utilized by other topic modeling approaches, including for example the RecoverKL algorithm of Arora et al. (2012). In the following the model geometry is exploited to derive fast and effective geometric algorithm for inference and parameter estimation.

# 3 Geometric inference of topics

We shall introduce a geometric loss function that can be viewed as a surrogate to the LDA's likelihood. To begin, let $\boldsymbol{\beta}$ denote the $K \times V$ topic matrix with rows $\beta_k$, $\boldsymbol{\theta}$ be a $M \times K$ document topic proportions matrix with rows $\theta_m$, and $\overline{W}$ be $M \times V$ normalized word counts matrix with rows $\bar{w}_m = w_m / N_m$.

## 3.1 Geometric surrogate loss to the likelihood

Unlike the original LDA formulation, here the Dirichlet distribution on $\boldsymbol{\theta}$ can be viewed as a prior on parameters $\boldsymbol{\theta}$. The log-likelihood of the observed corpora of $M$ documents is

$$L(\boldsymbol{\theta}, \boldsymbol{\beta}) = \sum_{m=1}^{M} \sum_{i=1}^{V} w_{mi} \log \left( \sum_{k=1}^{K} \theta_{mk} \beta_{ki} \right),$$

where the parameters $\boldsymbol{\beta}$ and $\boldsymbol{\theta}$ are subject to constraints $\sum_i \beta_{ki} = 1$ for each $k = 1, \ldots, K$, and $\sum_k \theta_{mk} = 1$ for each $m = 1, \ldots, M$. Partially relaxing these constraints and keeping only the one that the sum of all entries for each row of the matrix product $\boldsymbol{\theta}\boldsymbol{\beta}$ is 1, yields the upper bound that $L(\boldsymbol{\theta}, \boldsymbol{\beta}) \leq L(\overline{W})$, where function $L(\overline{W})$ is given by

$$L(\overline{W}) = \sum_m \sum_i w_{mi} \log \bar{w}_{mi}.$$

We can establish a tighter bound, which will prove useful (the proof of this and other technical results are in the Supplement):

**Proposition 1.** Given a fixed topic polytope $B$ and $\boldsymbol{\theta}$. Let $U_m$ be the set of words present in document $m$, and assume that $p_{mi} > 0 \, \forall \, i \in U_m$, then

$$L(\overline{W}) - \frac{1}{2} \sum_{m=1}^{M} N_m \sum_{i \in U_m} (\bar{w}_{mi} - p_{mi})^2 \geq L(\boldsymbol{\theta}, \boldsymbol{\beta}) \geq L(\overline{W}) - \sum_{m=1}^{M} N_m \sum_{i \in U_m} \frac{1}{p_{mi}} (\bar{w}_{mi} - p_{mi})^2.$$

Since $L(\overline{W})$ is constant, the proposition above shows that maximizing the likelihood has the effect of minimizing the following quantity with respect to both $\boldsymbol{\theta}$ and $\boldsymbol{\beta}$:

$$\sum_m N_m \sum_i (\bar{w}_{mi} - p_{mi})^2.$$

For each fixed $\boldsymbol{\beta}$ (and thus $B$), minimizing first with respect to $\boldsymbol{\theta}$ leads to the following

$$G(B) \quad := \quad \min_{\boldsymbol{\theta}} \sum_m N_m \sum_i (\bar{w}_{mi} - p_{mi})^2 = \sum_{m=1}^{M} N_m \min_{x : x \in B} \|x - \bar{w}_m\|_2^2, \tag{3}$$

where the second equality in the above display is due $p_m = \sum_k \theta_{mk} \beta_k \in B$. The proposition suggests a strategy for parameter estimation: $\boldsymbol{\beta}$ (and $B$) can be estimated by minimizing the geometric loss function $G$:

$$\min_B G(B) = \min_B \sum_{m=1}^{M} N_m \min_{x : x \in B} \|x - \bar{w}_m\|_2^2. \tag{4}$$

In words, we aim to find a convex polytope $B \in \Delta^{V-1}$, which is closest to the normalized word counts $\bar{w}_m$ of the observed documents. It is interesting to note the presence of document length $N_m$, which provides the weight for the squared $\ell_2$ error for each document. Thus, our loss function adapts to the varying length of documents in the collection. Without the weights, our objective is similar to the sum of squared errors of the Nonnegative Matrix Factorization(NMF). Ding et al. (2006) studied

the relation between the likelihood function of interest and NMF, but with a different objective of the NMF problem and without geometric considerations. Once $\hat{B}$ is solved, $\hat{\boldsymbol{\theta}}$ can be obtained as the barycentric coordinates of the projection of $\bar{w}_m$ onto $\hat{B}$ for each document $m = 1, \ldots, M$ (cf. Eq (3)). We note that if $K \leq V$, then $B$ is a simplex and $\beta_1, \ldots, \beta_k$ in general positions are the extreme points of $B$, and the barycentric coordinates are unique. (If $K > V$, the uniqueness no longer holds). Finally, $\hat{p}_m = \hat{\theta}_m^T \hat{\boldsymbol{\beta}}$ gives the cartesian coordinates of a point in $B$ that minimizes Euclidean distance to the maximum likelihood estimate: $\hat{p}_m = \underset{x \in B}{\operatorname{argmin}} \|x - \bar{w}_m\|_2$. This projection is not available in the closed form, but a fast algorithm is available (Golubitsky et al., 2012), which can easily be extended to find the corresponding distance and to evaluate our geometric objective.

### 3.2 Geometric Dirichlet Means algorithm

We proceed to devise a procedure for approximately solving the topic polytope $B$ via Eq. (4): first, obtain an estimate of the underlying subspace based on weighted k-means clustering and then, estimate the vertices of the polytope that lie on the subspace just obtained via a geometric correction technique. Please refer to the Supplement for a clarification of the concrete connection between our geometric loss function and other objectives which arise in subspace learning and weighted k-means clustering literature, the connection that motivates the first step of our algorithm.

**Geometric Dirichlet Means (GDM) algorithm**  estimates a topic polytope $B$ based on the training documents (see Algorithm 1). The algorithm is conceptually simple, and consists of two main steps: First, we perform a (weighted) k-means clustering on the $M$ points $\bar{w}_1, \ldots, \bar{w}_M$ to obtain the $K$ centroids $\mu_1, \ldots, \mu_K$, and second, construct a ray emanating from a (weighted) center of the polytope and extending through each of the centroids $\mu_k$ until it intersects with a sphere of radius $R_k$ or with the simplex $\Delta^{V-1}$ (whichever comes first). The intersection point will be our estimate for vertices $\beta_k$, $k = 1, \ldots, K$ of the polytope $B$. The center $C$ of the sphere is given in step 1 of the algorithm, while $R_k = \underset{1 \leq m \leq M}{\max} \|C - \bar{w}_m\|_2$, where the maximum is taken over those documents $m$ that are clustered with label $k$. To see the intuition behind the algorithm, let us consider a simple

---

**Algorithm 1** Geometric Dirichlet Means (GDM)

---

**Input:** documents $w_1, \ldots, w_M$, $K$,
    extension scalar parameters $m_1, \ldots, m_K$
**Output:** topics $\beta_1, \ldots, \beta_K$
  1: $C = \frac{1}{M} \sum_m \bar{w}_m$ {find center of the data}
  2: $\mu_1, \ldots, \mu_K =$ weighted k-means$(\bar{w}_1, \ldots, \bar{w}_M, K)$ {find centers of $K$ clusters}.
  3: **for all** $k = 1, \ldots, K$ **do**
  4:     $\beta_k = C + m_k (\mu_k - C)$.
  5:     **if** any $\beta_{ki} < 0$ **then** {threshold topic if it is outside vocabulary simplex $\Delta^{V-1}$}
  6:        **for all** $i = 1, \ldots, V$ **do**
  7:           $\beta_{ki} = \frac{\beta_{ik} \mathbb{1}_{\beta_{ki}>0}}{\sum_i \beta_{ki} \mathbb{1}_{\beta_{ki}>0}}$.
  8:        **end for**
  9:     **end if**
10: **end for**
11: $\beta_1, \ldots, \beta_K$.

---

simulation experiment. We use the LDA data generative model with $\alpha = 0.1$, $\eta = 0.1$, $V = 5$, $K = 4$, $M = 5000$, $N_m = 100$. Multidimensional scaling is used for visualization (Fig. 1). We observe that the k-means centroids (pink) do not represent the topics very well, but our geometric modification finds extreme points of the tetrahedron: red and yellow spheres overlap, meaning we found the true topics. In this example, we have used a very small vocabulary size, but in practice $V$ is much higher and the cluster centroids are often on the boundary of the vocabulary simplex, therefore we have to threshold the betas at 0. Extending length until $R_k$ is our default choice for the extension parameters:

$$m_k = \frac{R_k}{\|C - \mu_k\|_2} \text{ for } k = 1, \ldots, K, \tag{5}$$

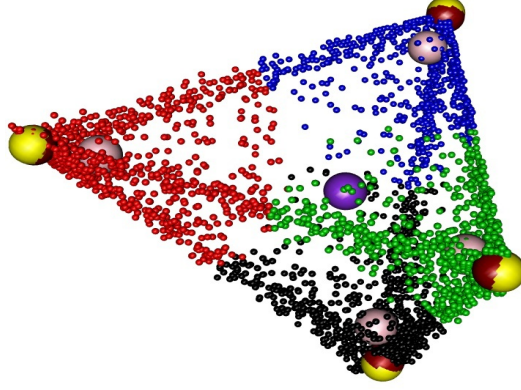

Figure 1: Visualization of GDM: Black, green, red and blue are cluster assignments; purple is the center, pink are cluster centroids, dark red are estimated topics and yellow are the true topics.

but we will see in our experiments that a careful tuning of the extension parameters based on optimizing the geometric objective (4) over a small range of $m_k$ helps to improve the performance considerably. We call this **tGDM** algorithm (tuning details are presented in the Supplement). The connection between extension parameters and the thresholding is the following: if the cluster centroid assigns probability to a word smaller than the whole data does on average, this word will be excluded from topic $k$ with large enough $m_k$. Therefore, the extension parameters can as well be used to control for the sparsity of the inferred topics.

### 3.3 Consistency of Geometric Dirichlet Means

We shall present a theorem which provides a theoretical justification for the Geometric Dirichlet Means algorithm. In particular, we will show that the algorithm can achieve consistent estimates of the topic polytope, under conditions on the parameters of the Dirichlet distribution of the topic proportion vector $\theta_m$, along with conditions on the geometry of the convex polytope $B$. The problem of estimating vertices of a convex polytope given data drawn from the interior of the polytope has long been a subject of convex geometry — the usual setting in this literature is to assume the uniform distribution for the data sample. Our setting is somewhat more general — the distribution of the points inside the polytope will be driven by a symmetric Dirichlet distribution setting, i.e., $\theta_m \overset{iid}{\sim} \mathrm{Dir}_K(\alpha)$. (If $\alpha = 1$ this results in the uniform distribution on $B$.) Let $n = K - 1$. Assume that the document multinomial parameters $p_1, \ldots, p_M$ (given in Eq. (2)) are the actual data. Now we formulate a geometric problem linking the population version of k-means and polytope estimation:

**Problem 1.** Given a convex polytope $A \in \mathbb{R}^n$, a continuous probability density function $f(x)$ supported by $A$, find a $K$-partition $A = \bigsqcup_{k=1}^{K} A_k$ that minimizes:

$$\sum_k^K \int_{A_k} \|\mu_k - x\|_2^2 f(x)\, dx,$$

where $\mu_k$ is the center of mass of $A_k$: $\mu_k := \frac{1}{\int_{A_k} f(x)\, dx} \int_{A_k} x f(x)\, dx$.

This problem is closely related to the Centroidal Voronoi Tessellations (Du et al., 1999). This connection can be exploited to show that

**Lemma 1.** Problem 1 has a unique global minimizer.

In the following lemma, a median of a simplex is a line segment joining a vertex of a simplex with the centroid of the opposite face.

**Lemma 2.** If $A \in \mathbb{R}^n$ is an equilateral simplex with symmetric Dirichlet density $f$ parameterized by $\alpha$, then the optimal centers of mass of the Problem 1 lie on the corresponding medians of $A$.

Based upon these two lemmas, consistency is established under two distinct asymptotic regimes.

**Theorem 1.** Let $B = \text{Conv}(\beta_1, \ldots, \beta_K)$ be the true convex polytope from which the $M$-sample $p_1, \ldots, p_M \in \Delta^{V-1}$ are drawn via Eq. (2), where $\theta_m \overset{iid}{\sim} \text{Dir}_K(\alpha)$ for $m = 1, \ldots, M$.

(a) If $B$ is also an equilateral simplex, then topic estimates obtained by the GDM algorithm using the extension parameters given in Eq. (5) converge to the vertices of $B$ in probability, as $\alpha$ is fixed and $M \to \infty$.

(b) If $M$ is fixed, while $\alpha \to 0$ then the topic estimates obtained by the GDM also converge to the vertices of $B$ in probability.

### 3.4 nGDM: nonparametric geometric inference of topics

In practice, the number of topics $K$ may be unknown, necessitating a nonparametric probabilistic approach such as the well-known Hierarchical Dirichlet Process (HDP) (Teh et al., 2006). Our geometric approach can be easily extended to this situation. The objective (4) is now given by

$$\min_B G(B) = \min_B \sum_{m=1}^{M} N_m \min_{x \in B} \|x - \bar{w}_m\|_2^2 + \lambda|B|, \tag{6}$$

where $|B|$ denotes the number of extreme points of convex polytope $B = \text{Conv}(\beta_1, \ldots, \beta_K)$. Accordingly, our nGDM algorithm now consists of two steps: (i) solve a penalized and weighted $k$-means clustering to obtain the cluster centroids (e.g. using DP-means (Kulis & Jordan, 2012)); (ii) apply geometric correction for recovering the extreme points, which proceeds as before. Our theoretical analysis can be also extended to this nonparametric framework. We note that the penalty term is reminiscent of the DP-means algorithm of Kulis & Jordan (2012), which was derived under a small-variance asymptotics regime. For the HDP this corresponds to $\alpha \to 0$ — the regime in part (b) of Theorem 1. This is an unrealistic assumption in practice. Our geometric correction arguably enables the accounting of the non-vanishing variance in data. We perform a simulation experiment for varying values of $\alpha$ and show that nGDM outperforms the KL version of DP-means (Jiang et al., 2012) in terms of perplexity. This result is reported in the Supplement.

## 4 Performance evaluation

**Simulation experiments** We use the LDA model to simulate data and focus our attention on the perplexity of held-out data and minimum-matching Euclidean distance between the true and estimated topics (Tang et al., 2014). We explore settings with varying document lengths ($N_m$ increasing from 10 to 1400 - Fig. 2(a) and Fig. 3(a)), different number of documents ($M$ increasing from 100 to 7000 - Fig. 2(b) and Fig. 3(b)) and when lengths of documents are small, while number of documents is large ($N_m = 50$, $M$ ranging from 1000 to 15000 - Fig. 2(c) and Fig. 3(c)). This last setting is of particular interest, since it is the most challenging for our algorithm, which in theory works well given long documents, but this is not always the case in practice. We compare two versions of the Geometric Dirichlet Means algorithm: with tuned extension parameters (tGDM) and the default one (GDM) (cf. Eq. 5) against the **variational EM** (VEM) algorithm (Blei et al., 2003) (with tuned hyperparameters), **collapsed Gibbs sampling** (Griffiths & Steyvers, 2004) (with true data generating hyperparameters), and **RecoverKL** (Arora et al., 2012) and verify the theoretical upper bounds for topic polytope estimation (i.e. either $(\log M/M)^{0.5}$ or $(\log N_m/N_m)^{0.5}$) - cf. Tang et al. (2014) and Nguyen (2015). We are also interested in estimating each document's topic proportion via the projection technique. RecoverKL produced only a topic matrix, which is combined with our projection based estimates to compute the perplexity (Fig. 3). Unless otherwise specified, we set $\eta = 0.1$, $\alpha = 0.1$, $V = 1200$, $M = 1000$, $K = 5$; $N_m = 1000$ for each $m$; the number of held-out documents is 100; results are averaged over 5 repetitions. Since finding exact solution to the k-means objective is NP hard, we use the algorithm of Hartigan & Wong (1979) with 10 restarts and the k-means++ initialization. Our results show that (i) Gibbs sampling and tGDM have the best and almost identical performance in terms of statistical estimation; (ii) RecoverKL and GDM are the fastest while sharing comparable statistical accuracy; (iii) VEM is the worst in most scenarios due to its instability (i.e. often producing poor topic estimates); (iv) short document lengths (Fig. 2(c) and Fig. 3(c)) do not degrade performance of GDM, (this appears to be an effect of the law of large

numbers, as the algorithm relies on the cluster means, which are obtained by averaging over a large number of documents); (v) our procedure for estimating document topic proportions results in a good quality perplexity of the RecoverKL algorithm in all scenarios (Fig. 3) and could be potentially utilized by other algorithms. Additional simulation experiments are presented in the Supplement, which considers settings with varying $N_m$, $\alpha$ and the nonparametric extension.

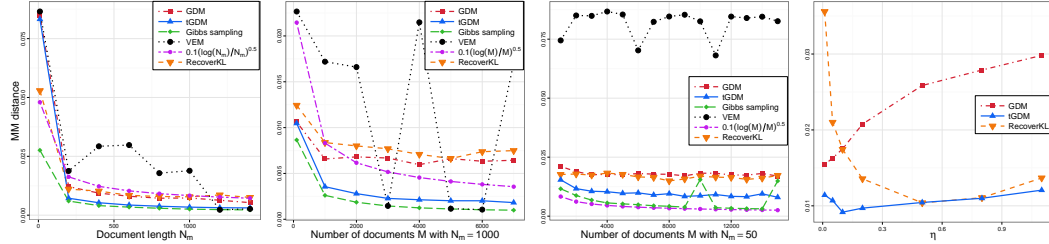

Figure 2: Minimum-matching Euclidean distance: increasing $N_m$, $M = 1000$ (a); increasing $M$, $N_m = 1000$ (b); increasing $M$, $N_m = 50$ (c); increasing $\eta$, $N_m = 50$, $M = 5000$ (d).

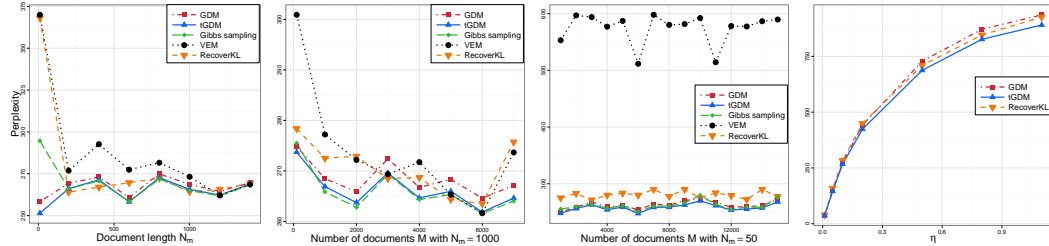

Figure 3: Perplexity of the held-out data: increasing $N_m$, $M = 1000$ (a); increasing $M$, $N_m = 1000$ (b); increasing $M$, $N_m = 50$ (c); increasing $\eta$, $N_m = 50$, $M = 5000$ (d).

**Comparison to RecoverKL**   Both tGDM and RecoverKL exploit the geometry of the model, but they rely on very different assumptions: RecoverKL requires the presence of anchor words in the topics and exploits this in a crucial way (Arora et al., 2012); our method relies on long documents in theory, even though the violation of this does not appear to degrade its performance in practice, as we have shown earlier. The comparisons are performed by varying the document length $N_m$, and varying the Dirichlet parameter $\eta$ (recall that $\beta_k|\eta \sim \text{Dir}_V(\eta)$). In terms of perplexity, RecoverKL, GDM and tGDM perform similarly (see Fig.4(c,d)), with a slight edge to tGDM. Pronounced differences come in the quality of topic's word distribution estimates. To give RecoverKL the advantage, we considered manually inserting anchor words for each topic generated, while keeping the document length short, $N_m = 50$ (Fig. 4(a,c)). We found that tGDM outperforms RecoverKL when $\eta \leq 0.3$, an arguably more common setting, while RecoverKL is more accurate when $\eta \geq 0.5$. However, if the presence of anchor words is not explicitly enforced, tGDM always outperforms RecoverKL in terms of topic distribution estimation accuracy for all $\eta$ (Fig. 2(d)). The superiority of tGDM persists even as $N_m$ varies from 50 to 10000 (Fig. 4(b)), while GDM is comparable to RecoverKL in this setting.

**NIPS corpora analysis**   We proceed with the analysis of the NIPS corpus.[1] After preprocessing, there are 1738 documents and 4188 unique words. Length of documents ranges from 39 to 1403 with mean of 272. We consider $K = 5, 10, 15, 20$, $\alpha = \frac{5}{K}$, $\eta = 0.1$. For each value of $K$ we set aside 300 documents chosen at random to compute the perplexity and average results over 3 repetitions. Our results are compared against Gibbs sampling, Variational EM and RecoverKL (Table 1). For $K = 10$, GDM with 1500 k-means iterations and 5 restarts in R took 50sec; Gibbs sampling with 5000 iterations took 10.5min; VEM with 750 variational, 1500 EM iterations and 3 restarts took 25.2min; RecoverKL coded in Python took 1.1min. We note that with recent developments (e.g.,

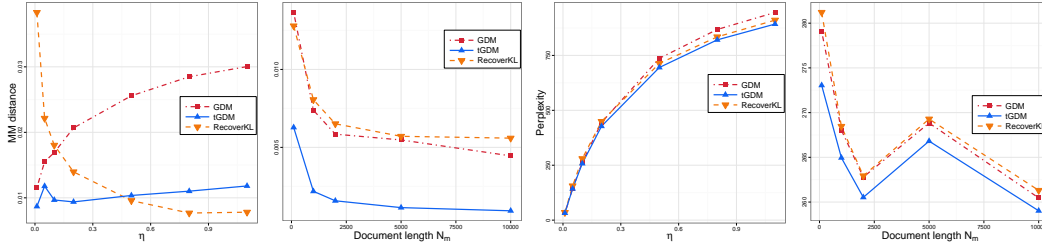

Figure 4: MM distance and Perplexity for varying $\eta$, $N_m = 50$ with anchors (a,c); varying $N_m$ (b,d).

(Hoffman et al., 2013)) VEM could be made faster, but its statistical accuracy remains poor. Although RecoverKL is as fast as GDM, its perplexity performance is poor and is getting worse with more topics, which we believe could be due to lack of anchor words in the data. We present topics found by Gibbs sampling, GDM and RecoverKL for $K = 10$ in the Supplement.

Table 1: Perplexities of the 4 topic modeling algorithms trained on the NIPS dataset.

|           | GDM  | RecoverKL | VEM  | Gibbs sampling |
|-----------|------|-----------|------|----------------|
| $K = 5$   | 1269 | 1378      | 1980 | 1168           |
| $K = 10$  | 1061 | 1235      | 1953 | 924            |
| $K = 15$  | 957  | 1409      | 1545 | 802            |
| $K = 20$  | 763  | 1586      | 1352 | 704            |

## 5 Discussion

We wish to highlight a conceptual aspect of GDM distinguishing it from moment-based methods such as RecoverKL. GDM operates on the document-to-document distance/similarity matrix, as opposed to the second-order word-to-word matrix. So, from an optimization viewpoint, our method can be viewed as the dual to RecoverKL method, which must require anchor-word assumption to be computationally feasible and theoretically justifiable. While the computational complexity of RecoverKL grows with the vocabulary size and not the corpora size, our convex geometric approach continues to be computationally feasible when number of documents is large: since only documents near the polytope boundary are relevant in the inference of the extreme points, we can discard most documents residing near the polytope's center.

We discuss some potential improvements and extensions next. The tGDM algorithm showed a superior performance when the extension parameters are optimized. This procedure, while computationally effective relative to methods such as Gibbs sampler, may still be not scalable to massive datasets. It seems possible to reformulate the geometric objective as a function of extension parameters, whose optimization can be performed more efficiently. In terms of theory, we would like to establish the error bounds by exploiting the connection of topic inference to the geometric problem of Centroidal Voronoi Tessellation of a convex polytope.

The geometric approach to topic modeling and inference may lend itself naturally to other LDA extensions, as we have demonstrated with nGDM algorithm for the HDP (Teh et al., 2006). Correlated topic models of Blei & Lafferty (2006a) also fit naturally into the geometric framework — we would need to adjust geometric modification to capture logistic normal distribution of topic proportions inside the topic polytope. Another interesting direction is to consider dynamic (Blei & Lafferty, 2006b) (extreme points of topic polytope evolving over time) and supervised (McAuliffe & Blei, 2008) settings. Such settings appear relatively more challenging, but they are worth pursuing further.

### Acknowledgments

This research is supported in part by grants NSF CAREER DMS-1351362 and NSF CNS-1409303.

## Footnotes

[1] https://archive.ics.uci.edu/ml/datasets/Bag+of+Words

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
