[Supplementary Material]

# Supplementary material for Geometric Dirichlet Means algorithm for topic inference

**Mikhail Yurochkin**
Department of Statistics
University of Michigan
moonfolk@umich.edu

**XuanLong Nguyen**
Department of Statistics, Department of EECS
University of Michigan
xuanlong@umich.edu

## 1 Proof of Proposition 1

*Proof.* Consider the KL divergence between two distributions parameterized by $\bar{w}_m$ and $p_m$, respectively:

$$D(P_{\bar{w}_m} \| P_{p_m}) = \sum_{i \in U_m} \bar{w}_{mi} \log \frac{\bar{w}_{mi}}{p_{mi}}$$

$$= \frac{1}{N_m} \left( \sum_{i \in U_m} w_{mi} \log \bar{w}_{mi} - \sum_{i \in U_m} w_{mi} \log p_{mi} \right).$$

Then $L(\overline{W}) - L(\boldsymbol{\theta}, \boldsymbol{\beta}) = \sum_m N_m D(P_{\bar{w}_m} \| P_{p_m}) \geq 0$, due to the non-negativity of KL divergence. Now we shall appeal to a standard lower bound for the KL divergence (Cover & Joy, 2006):

$$D(P_{\bar{w}_m} \| P_{p_m}) \geq \frac{1}{2} \sum_{i \in U_m} (\bar{w}_{mi} - p_{mi})^2,$$

and an upper bound via $\chi^2$-distance (e.g. see Sayyareh (2011)):

$$D(P_{\bar{w}_m} \| P_{p_m}) \leq \sum_{i \in U_m} \frac{1}{p_{mi}} (\bar{w}_{mi} - p_{mi})^2.$$

Taking summation of both bounds over $m = 1, \dots, M$ concludes the proof. $\square$

## 2 Connection between our geometric loss function and other objectives which arise in subspace learning and k-means clustering problems.

Recall that our geometric objective is:

$$\min_B G(B) = \min_B \sum_{m=1}^{M} N_m \min_{x:x \in B} \|x - \bar{w}_m\|_2^2. \tag{1}$$

We note that this optimization problem can be reduced to two other well-known problems when the objective function and constraints are suitably relaxed/modified:

- A version of weighted low-rank matrix approximation is $\min_{\text{rank}(\hat{D}) \leq r} \text{tr}((\hat{D} - D)^T Q (\hat{D} - D))$.

  If $Q = \text{diag}(N_1, \dots, N_M)$, $D = \overline{W}$, $r = K$ and $\hat{D} = \boldsymbol{\theta}\boldsymbol{\beta}$, the problem looks similar to the geometric objective without constraints and has a closed form solution (Manton et al., 2003): $\hat{D} = Q^{-1/2} U \Sigma_K V^T$, where

$$Q^{1/2} D = U \Sigma_K V^T \tag{2}$$

is the singular value decomposition and $\Sigma_K$ is the truncation to $K$ biggest singular values. Also note that here and further without loss of generality we assume $M \geq V$, if $M < V$ for the proofs to hold we replace $Q^{1/2}D$ with $(Q^{1/2}\overline{W})^T$.

- The k-means algorithm involves optimizing the objective (Hartigan & Wong, 1979; Lloyd, 1982; MacQueen, 1967): $\min_{x_1,\ldots,x_K} \sum_m \min_{i \in \{1,\ldots,K\}} \|\bar{w}_m - x_i\|_2^2$. Our geometric objective (1) is quite similar — it replaces the second minimization with minimizing over the convex hull of $\{x_1, \ldots, x_K\}$ and includes weight $N_m$s.

- The two problems described above are connected in the following way (Xu et al., 2003). Define the weighted k-means objective with respect to cluster assignments: $\sum_k \sum_{m \in C_k} N_m \|\bar{w}_m - \mu_k\|^2$, where $\mu_k$ is the centroid of the $k$-th cluster:

$$\mu_k = \frac{\sum_{m \in C_k} N_m \bar{w}_m}{\sum_{m \in C_k} N_m}. \tag{3}$$

Let $S_k$ be the optimal indicator vector of cluster $k$, i.e., $m$-th element is 1 if $m \in C_k$ and 0 otherwise. Define

$$Y_k = \frac{Q^{1/2} S_k}{\|Q^{1/2} S_k\|_F^2}. \tag{4}$$

If we relax the constraint on $S_k$ to allow any real values instead of only binary values, then $Y$ can be solved via the following eigenproblem: $Q^{1/2}\overline{W}\,\overline{W}^T Q^{1/2} Y = \lambda Y$.

Let us summarize the above observations by the following:

**Proposition 2.** Given the $M \times V$ normalized word counts matrix $\overline{W}$. Let $\mu_1, \ldots, \mu_K$ be the optimal cluster centroids of the weighted k-means problem given by Eq. (3), and let $v_k$s be the columns of $V$ in the SVD of Eq. (2). Then,

$$\mathrm{span}(\mu_1, \ldots, \mu_K) = \mathrm{span}(v_1, \ldots, v_K).$$

*Proof.* Following Ding & He (2004), let $P_c$ be an operator projecting any vector onto $\mathrm{span}(\mu_1, \ldots, \mu_K)$: $P_c = \sum_k \mu_k \mu_k^T$. Recall that $S_k$ is the indicator vector of cluster $k$ and $Y_k$ defined in Eq. (4). Then $\mu_k = \frac{\overline{W}^T Q S_k}{\|Q^{1/2} S_k\|_F^2} = \overline{W}^T Q^{1/2} Y_k$, and $P_c = \sum_k \overline{W}^T Q^{1/2} Y_k (\overline{W}^T Q^{1/2} Y_k)^T$. Now, note that $Y_k$'s are the eigenvectors of $Q^{1/2}\overline{W}\,\overline{W}^T Q^{1/2}$, which are also left-singular vectors of $Q^{1/2}\overline{W} = U\Sigma V^T$, so

$$P_c = (Q^{1/2}\overline{W})^T Y_k ((Q^{1/2}\overline{W})^T Y_k)^T = \sum_k \lambda_k^2 v_k v_k^T,$$

which is the projection operator for $\mathrm{span}(v_1, \ldots, v_K)$. Hence, the two subspaces are equal. $\square$

Prop. 2 and the preceding discussions motivate the GDM algorithm for estimating the topic polytope: first, obtain an estimate of the underlying subspace based on k-means clustering and then, estimate the vertices of the polytope that lie on the subspace just obtained.

## 3  Proofs of technical lemmas

Recall from the main part:

**Problem 1.** Given a convex polytope $A \in \mathbb{R}^n$, a continuous probability density function $f(x)$ supported by $A$, find a $K$-partition $A = \bigsqcup_{k=1}^{K} A_k$ that minimizes:

$$\sum_{k}^{K} \int_{A_k} \|\mu_k - x\|_2^2 f(x) \, dx,$$

where $\mu_k$ is the center of mass of $A_k$: $\mu_k := \frac{1}{\int_{A_k} f(x) \, dx} \int_{A_k} x f(x) \, dx$.

**Proof of Lemma 1**

*Proof.* The proof follows from a sequence of results of Du et al. (1999), which we now summarize. First, if the $K$-partition $(A_1, \ldots, A_K)$ is a minimizer of Problem 1, then $A_k$s are the Voronoi regions corresponding to the $\mu_k$s. Second, Problem 1 can be restated in terms of the $\mu_k$s to minimize $\mathcal{K}(\mu_1, \ldots, \mu_K) = \sum_k \int_{\hat{A}_k} \|\mu_k - x\|_2^2 f(x) \, dx$, where $\hat{A}_k$s are the Voronoi regions corresponding to their centers of mass $\mu_k$s. Third, $\mathcal{K}(\mu_1, \ldots, \mu_K)$ is a continuous function and admits a global minimum. Fourth, the global minimum is unique if the distance function in $\mathcal{K}$ is strictly convex and the Voronoi regions are convex. Now, it can be verified that the squared Euclidean distance is strictly convex. Moreover, Voronoi regions are intersections of half-spaces with the convex polytope $A$, which can also be represented as an intersection of half-spaces. Therefore, the Voronoi regions of Problem 1 are convex polytopes, and it follows that the global minimizer is unique. □

**Proof of Lemma 2**

*Proof.* Since $f$ is a symmetric Dirichlet density, the center of mass of $A$ coincides with its centroid. Let $n = 3$. In an equilateral triangle, the centers of mass $\mu_1, \mu_2, \mu_3$ form an equilateral triangle $C$. An intersection point of the Voronoi regions $A_1, A_2, A_3$ is the circumcenter and the centroid of $C$, which is also a circumcenter and centroid of $A$. Therefore, $\mu_1, \mu_2, \mu_3$ are located on the medians of $A$ with exact positions depending on the $\alpha$. The symmetry and the property of circumcenter coinciding with centroid carry over to the general $n$-dimensional equilateral simplex (Westendorp, 2013). □

## 4   Proof of consistency theorem

*Proof.* For part (a), let $(\hat{\mu}_1, \ldots, \hat{\mu}_K)$ be the minimizer of the k-means problem $\min_{\mu_1, \ldots, \mu_K} \sum_m \min_{i \in \{1, \ldots, K\}} \|p_m - \mu_i\|_2^2$. Let $\tilde{\mu}_1, \ldots, \tilde{\mu}_K$ be the centers of mass of the solution of Problem 1 applied to $B$ and the Dirichlet density. By Lemma 1, these centers of mass are unique, as they correspond to the unique optimal $K$-partition. Accordingly, by the strong consistency of k-means clustering under the uniqueness condition (Pollard, 1981), as $M \to \infty$,

$$\text{Conv}(\hat{\mu}_1, \ldots, \hat{\mu}_K) \to \text{Conv}(\tilde{\mu}_1, \ldots, \tilde{\mu}_K) \text{ a.s.},$$

where the convergence is assessed in either Hausdorff or the minimum matching distance for convex sets (Nguyen, 2015). Note that $C = \frac{1}{M} \sum_m p_m$ is a strongly consistent estimate of the centroid $C_0$ of $B$, by the strong law of large numbers. Lemma 2 shows that $\tilde{\mu}_1, \ldots, \tilde{\mu}_K$ are located on the corresponding medians. To complete the proof, it remains to show that $\hat{R} := \max_{1 \leq m \leq M} \|C - p_m\|_2$ is a weakly consistent estimate of the circumradius $R_0$ of $B$. Indeed, for a small $\epsilon > 0$ define the event $E_m^k = \{p_m \in B_\epsilon(\beta_k) \cap B\}$, where $B_\epsilon(\beta_k)$ is an $\epsilon$-ball centering at vertex $\beta_k$. Since $B$ is equilateral and the density over it is symmetric and positive everywhere in the domain, $\mathbb{P}(E_m^1) = \ldots = \mathbb{P}(E_m^K) =: b_\epsilon > 0$. Let $E_m = \bigcup_k E_m^k$, then $\mathbb{P}(E_m) = b_\epsilon K$. We have

$$\limsup_{M \to \infty} \mathbb{P}(|\hat{R} - R_0| > 2\epsilon) = \limsup_{M \to \infty} \mathbb{P}(\max_{1 \leq m \leq M} \|C_0 - p_m\|_2 < R_0 - \epsilon) <$$

$$< \limsup_{M \to \infty} \mathbb{P}(\bigcap_{m=1}^{M} E_m^{\complement}) = \limsup_{M \to \infty} (1 - b_\epsilon K)^M = 0.$$

A similar argument allows us to establish that each $R_k$ is also a weakly consistent estimate of $R_0$. This completes the proof of part (a). For a proof sketch of part (b), for each $\alpha > 0$, let $(\mu_1^\alpha, \ldots, \mu_K^\alpha)$ denote the $K$ means obtained by the k-means clustering algorithm. It suffices to show that these estimates converge to the vertices of $B$. Suppose this is not the case, due to the compactness of $B$, there is a subsequence of the $K$ means, as $\alpha \to 0$, that tends to $K$ limit points, some of which are not the vertices of $B$. It is a standard fact of Dirichlet distributions that as $\alpha \to 0$, the distribution of the $p_m$ converges weakly to the discrete probability measure $\sum_{k=1}^{K} \frac{1}{K} \delta_{\beta_k}$. So the k-means objective function tends to $\frac{M}{K} \sum_k \min_{i \in \{1, \ldots, K\}} \|\beta_k - \mu_i^\alpha\|_2^2$, which is strictly bounded away from 0, leading to a contradiction. This concludes the proof. □

## 5  Tuned GDM

In this section we discuss details of the extension parameters tuning. Recall that GDM requires extension scalar parameters $m_1, \ldots, m_K$ as part of its input. Our default choice is

$$m_k = \frac{R_k}{\|C - \mu_k\|_2} \text{ for } k = 1, \ldots, K, \tag{5}$$

where $R_k = \max_{m \in C_k} \|C - \bar{w}_m\|_2$ and $C_k$ is the set of indices of documents belonging to cluster $k$. In some situations (e.g. outliers making extension parameters too big) tuning of the extension parameters can help to improve the performance, which we called **tGDM** algorithm. Recall the geometric objective (1) and let

$$G_k(B) := \sum_{m \in C_k} N_m \min_{x : x \in B} \|x - \bar{w}_m\|_2^2, \tag{6}$$

which is simply the geometric objective evaluated at the documents of cluster $k$. For each $k = 1, \ldots, K$ we used line search procedure (Brent, 2013) optimization of $G_k(B)$ in an interval from 1 up to default $m_k$ as in (5). Independent tuning for each $k$ gives an approximate solution, but helps to reduce the running time.

## 6  Performance evaluation

Here we present some additional simulation results and NIPS topics.

**Nonparametric analysis with DP-means.**  Based on simulations we show how nGDM can be used when number of topics is unknown and compare it against DP-means utilizing KL divergence (KL DP-means) by Jiang et al. (2012). We analyze settings with $\alpha$ ranging from 0.01 to 2. Recall that KL DP-means assumes $\alpha \to 0$. $V = 1200$, $M = 2000$, $N_m = 3000$, $\eta = 0.1$, true $K = 15$. For each value of $\alpha$ average over 5 repetitions is recorded and we plot the perplexity of 100 held-out documents. Fig. 1 supports our argument - for small values of $\alpha$ both methods perform equivalently well (KL DP-means due to variance assumption being satisfied and nGDM due to part (b) of Theorem 1), but as $\alpha$ gets bigger, we see how our geometric correction leads to improved performance.

Figure 1: Perplexity for varying $\alpha$

**Documents of varying size.**  Until this point all documents are of the same length. Next, we evaluate the improvement of our method when document length varies. The lengths are randomly sampled from 50 to 1500 and the experiment is repeated 20 times. The weighted GDM uses document lengths as weights for computing the data center and training k-means. In both performance measures (Fig. 2 left and center) the weighted version consistently outperforms the unweighted one, while the tuned weighted version stays very close to Gibbs sampling results.

**Effect of the document topic proportions prior.** Recall that topic proportions are sampled from the Dirichlet distribution $\theta_m | \alpha \sim \text{Dir}_K(\alpha)$. We let $\alpha$ increase from 0.01 to 2. Smaller $\alpha$ implies that samples are close to the extreme points, and hence GDM estimates topics better. This also follows from Theorem 1(b) of the paper. We see (Fig. 2 right) that our solution and Gibbs sampling are almost identical for small $\alpha$, while VEM is unstable. With increased $\alpha$ Gibbs sampling remains the best, while our algorithm remains better than VEM. We also note that increasing $\alpha$ causes error of all methods to increase.

Figure 2: Minimum-matching Euclidean distance: varying $N_m$ (left); increasing $\alpha$ (right). Perplexity for varying $N_m$ (center).

**Projection estimate analysis.** Our objective function (1) motivates the estimation of document topic proportions by taking the barycentric coordinates of the projection of the normalized word counts of a document onto the topic polytope. To do this we utilized the projection algorithm of Golubitsky et al. (2012). Note that some algorithms (RecoverKL in particular) do not have a built in method for finding topic proportions of the unseen documents. Our projection based estimate can solve this issue, as it can find topic proportions of a document only based on the topic polytope. Fig. 3 shows that perplexity with projection estimates closely follows corresponding results and outperforms VEM on the short documents (Fig. 3 (right)).

Figure 3: Projection method: increasing $N_m$, $M = 1000$ (left); increasing $M$, $N_m = 1000$ (center); increasing $M$, $N_m = 50$ (right).

**Top 10 words (columns) of each of the 10 learned topics of NIPS dataset**

| GDM topics | | | | | | | | | |
|---|---|---|---|---|---|---|---|---|---|
| analog | regress. | reinforc. | nodes | speech | image | mixture | neurons | energy | rules |
| circuit | kernel | policy | node | word | images | experts | neuron | characters | teacher |
| memory | bayesian | action | classifier | hmm | object | missing | cells | boltzmann | student |
| chip | loss | controller | classifiers | markov | visual | mixtures | cell | character | fuzzy |
| theorem | posterior | actions | tree | phonetic | objects | expert | synaptic | hopfield | symbolic |
| sources | theorem | qlearning | trees | speaker | face | gating | spike | temperature | saad |
| polynom. | hyperp. | reward | bayes | acoustic | pixel | posterior | activity | annealing | membership |
| separation | bounds | sutton | rbf | phoneme | pixels | tresp | firing | kanji | rulebased |
| recurrent | monte | robot | theorem | hmms | texture | loglikel. | visual | adjoint | overlaps |
| circuits | carlo | barto | boolean | hybrid | motion | ahmad | cortex | window | children |

| Gibbs sampler topics | | | | | | | | | |
|---|---|---|---|---|---|---|---|---|---|
| neurons | rules | mixture | reinforc. | memory | speech | image | analog | theorem | classifier |
| cells | language | bayesian | policy | energy | word | images | circuit | regress. | nodes |
| cell | recurrent | posterior | action | neurons | hmm | visual | chip | kernel | node |
| neuron | node | experts | robot | neuron | auditory | object | voltage | loss | classifiers |
| activity | tree | entropy | motor | capacity | sound | motion | neuron | bounds | tree |
| synaptic | memory | mixtures | actions | hopfield | phoneme | objects | vlsi | proof | clustering |
| firing | nodes | markov | controller | associative | acoustic | spatial | circuits | polynom. | character |
| spike | symbol | separation | trajectory | recurrent | hmms | face | digital | lemma | rbf |
| stimulus | symbols | sources | arm | attractor | mlp | pixel | synapse | teacher | cluster |
| cortex | grammar | principal | reward | boltzmann | segment. | pixels | gate | risk | characters |

| RecoverKL topics | | | | | | | | | |
|---|---|---|---|---|---|---|---|---|---|
| entropy | reinforc. | classifier | loss | ensemble | neurons | penalty | mixture | validation | image |
| image | controller | classifiers | theorem | energy | neuron | rules | missing | regress. | visual |
| kernel | policy | speech | bounds | posterior | spike | regress. | recurrent | bayesian | motion |
| energy | action | nodes | proof | bayesian | synaptic | bayesian | bayesian | crossvalid. | cells |
| ica | actions | word | lemma | speech | cells | energy | posterior | risk | neurons |
| images | memory | node | polynom. | boltzmann | firing | theorem | image | stopping | images |
| separation | robot | image | neurons | student | cell | analog | markov | tangent | receptive |
| clustering | trajectory | tree | regress. | face | activity | regulariz. | speech | image | circuit |
| sources | sutton | character | nodes | committee | synapses | recurrent | images | kernel | spatial |
| mixture | feedback | memory | neuron | momentum | stimulus | perturb. | object | regulariz. | object |