[Reviews · NeurIPS 2016]

Reviewer 1

Summary

The paper provides a geometric perspective on the topic inference problem for LDA. By integrating out the latent topic labels for individual words and instead focusing on the multinomial distribution of the word-count vectors of the documents given the word-topic distributions beta and the document topic distributions theta as parameters, the paper shows that maximizing likelihood under the LDA is equivalent to minimizing a geometric loss function. The optimization problem amounts to identifying a simplex of topic distributions that is close to all document word vectors generated from it in an L2 sense from all symmetric Dirichlet distributions over the topic simplex. To solve this problem, the paper makes a connection to the problem of centroidal voronoi tesselations of a convex polytope and the k-means algorithm. Based on this connection, the geometric dirichlet means algorithm for LDA first uses the k-means algorithm to make initial estimates of the topics. And then it makes a geometric correction by projecting the k-means estimates along the medians of the topic simplex to ensure that they lie inside the vocabulary simplex and within a high-probability sphere. The algorithm is shown to be consistent under regularity conditions involving the Dirichlet parameters and geometry of the topic simplex. Experimental results show that this geometric algorithm produces topics that are as accurate / generalizable as those from Gibbs Sampling and the algorithm is as fast as the recent RecoverKL algorithm.

Qualitative Assessment

I like this paper for two different reasons. After RecoverKL and the spectral algorithm, this paper brings a very novel and useful perspective into the topic inference problem for LDA, without apparently making strong assumptions about topics, such as separability via anchor words, etc. Secondly, it seems to be extremely good in practice meeting the speed of RecoverKL with the accuracy of Gibbs sampling algorithms. A. The algorithm: Aspects of this work were known before. For example, Blei pointed out the convex geometry in the original LDA paper, and the connection between LDA/NMF and K-Means was also known. However, the novel aspect of this paper is that it has used these connections to propose an inference algorithm for LDA completely based on the geometry of the topic and word simplexes. This is done by making an additional connection between the topic inference problem and that of Centroidal Voronoi Tesselations of a convex simplex. The algorithm is simple at a high level: find initial topics by running K-Means on the document word counts, and then make a geometric correction by projecting them along the medians of the simplex to intersect either the vocabulary simplex or a high probability sphere. Yet, it is guaranteed to recover the true topics under some assumptions about the geometry of the topic simplex and vocab simplexes and the Dirichlet distribution. B. Evaluations: The evaluations using synthetic and real data are quite detailed and show that the new perspective is useful as well. Based on the evidence presented, the proposed algorithm seems to be the best out there for LDA in terms of topic recovery accuracy and execution time. The comparisons against RecoverKL are also quite detailed with allowances made for the anchor-word assumption of RecoverKL. The following are some specific questions / thoughts that I had while reading the paper. 1. The implication of the regularity conditions: The regularity conditions are not interpreted clearly enough in the paper. The second regime with alpha -> 0 essentially means degeneration of an admixture model (where documents lie in the interior of the topic simplex) to a mixture model (where documents lie on the vertices.) What does an equilateral topic simplex mean? All topics are equally distant from each other in an L2 sense? 2. Intuition behind the algorithm: Understanding the motivation behind the actual algorithm without reading the supplementary material was hard. The intuition behind the geometry is brought out by the associated explanation and the figure but not the intuition behind the algorithm. It is not explained that the "ray" is in fact along the median, assuming that the initial mean is already on the median. Here and in section 3.3 it is important to distinguish between the two different simplexes that are of importance - the vocab simplex and the topic simplex. 4. Performance of RecoverKL: I found it surprising that while the perplexity pattern for all the other algorithms is decreasing with topics, the curve for RecoverKL is different. What is the explanation for this? 5. Identifiability. The paper only talks about identifiability of the convex-hull of the topic polytope and not that of the topic polytope itself. This point is not made clearly enough in the paper.

Confidence in this Review

2-Confident (read it all; understood it all reasonably well)


Reviewer 2

Summary

This paper proposes a new inference algorithm fo

Qualitative Assessment

This paper proposes a new inference algorithm based on a geometric loss function. In the experiments, the proposed method achieves comparable perplexity with Gibbs sampling with less computational time. I have concern about the experimental setting. With the experiments, hyperparameters, \alpha and \eta, are fixed. However, with VEM and Gibbs sampling, the hyperparameters are usually estimated and it is known that estimating hyperparameters improves their perplexity. I would like to see the perplexity when the hyperparamters are estimated with VEM and Gibbs sampling. With the proposed method, how the hyperparameters are used? In the Algorithm1, the hyperparameters are not appeared. The proposed method does not use the hyperparameters? If the proposed method uses the hyperparameters, is it possible to estimate them from the given data? p.1: contain -> contains

Confidence in this Review

1-Less confident (might not have understood significant parts)


Reviewer 3

Summary

The paper presents a geometric viewpoint to the training of LDA and a corresponding training algorithm. The algorithm is compared against several other important training algorithms for LDA: Gibbs sampling, variational inference, recoverKL.

Qualitative Assessment

I really enjoyed this paper. It is very original and full of insight. This is not the first paper to mention the geometric aspects to LDA, but I had not seen before a paper that solves the training of LDA using it. I found it to be a hard paper because it is dense with information, but thankfully it is well written and clear. I think this paper could have a significant impact because it exemplifies how considering the geometry of a probabilistic model might lead to new and potentially scalable inference algorithms. Considering that this is a very original algorithm, I was satisfied with the experimental evaluation. The dataset and the number of topics is small, but many details about the experiment are reported, and the proposed algorithm is compared against the three major other training algorithms for LDA. I think the experimental evaluation is good enough for more people to be interested in this approach and decide to investigate further and repeat the experiment. In section 3.1, I assume that the min within the big summation is binding variable x. 1. If not, I really misunderstood something and more explanations would be welcome. 2. If it is binding x, this notation is somewhat ambiguous. The transition from 3.1 to 3.2 is abrupt. It is explained throughout the paper in section 3.3 and in the appendix, but it would really help if you tried to explain how we went from the surrogate loss to GDM, and give some intuition about it.

Confidence in this Review

2-Confident (read it all; understood it all reasonably well)


Reviewer 4

Summary

The paper proposes a geometric view to topic modeling and develops an inference algorithm based on it. This is an interesting point of view and an inference method. The authors provide theoretical guarantees of their method proving a theorem about convergence of their topic estimates to the true ones. Topic modeling is considered as a geometric problem, where there is a convex hull of topics and documents, represented as vectors of words probabilities, are points inside this polytope. A topics proportions vector for a document can be then represented as barycentric coordinates of this document w.r.t. this polytope. The learning procedure consists on two steps: finding centroids of K-means clusters, where K is the number of topics, first, and then extending these centroids to the vertices of the polytope along rays that connect centroids and the barycenter of the polytope. Inference of topic proportions for documents is based on projection of documents points, represented as vectors of normalised words frequencies, to this found polytope. The algorithm is shown to be fast and accurate in terms of hold-out perplexity and minimum-matching Euclidean distance between true and estimated topics. Moreover, it seems to be possible to extend this approach for more complex topic models. The authors present the extension for the case where the number of topics is unknown and discuss possibility of extensions to correlated and dynamic topic models.

Qualitative Assessment

The paper reads well, despite the fact that a lot of information is presented in a dense form. The authors refer everywhere to related work. Although the paper is presented well, there are ways to improve it. There are two most crucial aspects which are currently not very well covered in the paper, everything else is not critical, they are some suggestions for improvement. These two crucial moments are descriptions of tuning of extension parameters in the proposed tGDM algorithm and the first step in the proposed extension of the algorithm called as the nGDM algorithm. tGDM algorithm: The authors claim that extension of the centroids of the k-means clusters towards vertices of the polytope by the fixed length (as it is in the basic proposed algorithm called as the GDM algorithm) is not always a good choice. They mention that "careful tuning of the extension parameters based on optimizing the geometric objective over a small range of m_k helps to improve the performance considerably". And they call this tGDM algorithm. It is not clear for me how exactly this tuning happens. It would be good to add some more details about it. nGDM algorithm: The authors present an extension of their approach to the case when the number of topics is unknown. They say that the first step of their algorithm "involves solving a penalized and weighted k-means clustering". It is unclear how it can be done. If it refers to some known method it should be cited, otherwise it would be also better to add some more details about this step. Other suggestions: 1. As the material is dense it would be good to summarise the whole algorithm with specific implementations of the steps 2. It looks strange that in sections 3.1 and 3.2 thetas are treated as parameters and then in section 3.3 they transform to be random variables. It is better, at least, to mention about this transformation 3. To continue the previous remark the likelihood in section 3.1 is for the PLSA model, rather than for the LDA one (because of treatment thetas as parameters rather than random variables). It would be better to state this and to clarify about LDA likelihood as the claim about connection between the geometric objective function and LDA likelihood appears several times in the paper 4. The claim "V is much higher and the cluster centroids are often on the boundary of the vocabulary simplex, therefore we have to threshold the betas at 0" (lines 146-148) requires some explanations 5. It would be better to add some citation to the claim in the sentence in lines 67-68 6. Regarding experiments. In the paper it is claimed that the results of the variational EM algorithm are always worse than the others but according to the results in figure 2 in the supplementary materials (where there are results for individual runs of the algorithms rather than averaged ones presented in the paper) for only 3 out of 20 different runs of the variational EM algorithm its results are worse than the results of Gibbs sampling and these results of both algorithms coincide in other 17 runs. It looks not so bad. Maybe averaging procedure of the results of the individual runs of the algorithms is not appropriate. At least it is worth to mention about this behaviour. It is mentioned in the paper that variational EM is unstable. Even if it regards to this behaviour it is better to explain it in more details 7. Information about Gibbs sampling and the variational EM algorithm settings in the experiments on simulated data would be good 8. In Figure 2 there are results referred as 0.1(log(N_m)/N_m)^0.5. There is no information about this in the text and it is unclear what it is about 9. The claim that tuning extension parameters could be computationally expensive on big datasets (lines 248-249) looks strange as there was no possibility to check it 10. It is not very clear why there are no results for Gibbs sampling and the variational EM algorithm in figures 2d and 3d 11. Regarding results presented in the supplementary materials. It is not clear for me what is weighted GDM. As I understand GDM presented in the paper is already weighted 12. It is not clear what the reference "projection with VEM" in figure 3 in the supplementary materials refers to 13. It is not clear why there is no presented topics inferred by the variational EM algorithm in the supplementary materials Minor things: 1. Please check style requirements, e.g. a paper title should not be capitalised 2. Acronyms GDM, VEM and MLE are undefined 3. It would be better to define the notation "Dir_V" 4. It would be better to distinguish scalars and vectors in notations 5. There is "for m = 1, ..., M" in line 58 but there is not "for k = 1, ..., K" in line 57 6. "z", "d" and "n_m" are never defined 7. B := ... in line 77 rather than B = ... 8. "lets" in line 80 seems to be "let" 9. Missing articles before "document" in lines 83 and 84 10. Please enumerate all equations 11. In equation between 107 and 108 V is used as a set, while it is introduced as a scalar 12. Missing article before "topic polytope" in line 133 13. "Generative" rather than "generating" in line 142 14. It seems that there is only one tetrahedron in the simple simulation experiment (lines 140-146), then "tetrahedra" in line 145 should be "tetrahedron" 15. It is better to add subcaptures a, b, c and d in figures 2 - 4 as they are referred. 16. Missing closing bracket in line 243 17. It is better to add citation to the used NIPS corpora 18. Please checked references: there are lower cases for proper names, e.g. "Hierarchical dirichlet processes"; there are also caps for articles, e.g. in "Proceedings of The 31st International ..."; papers from NIPS are referred as "NIPS, 2012" in line 283 and "Advances in neural information processing systems, 2006a" in line 286; authors' given names appear both in full forms and as initials; a place of a conference is stated only for one reference in line 307

Confidence in this Review

2-Confident (read it all; understood it all reasonably well)


Reviewer 5

Summary

In this paper authors study a geometric loss function, which can be viewed as a surrogate to the LDA’s likelihood. This leads to a novel estimation and inference algorithm named the Geometric Dirichlet Means (GDM) algorithm. The paper proves that the GDM algorithm is consistent, under conditions on the Dirichlet distribution and the geometry of the topic polytope.

Qualitative Assessment

The paper is well written and easy to follow. The proposed algorithm is novel and interesting. However, the experiments presented in Section 4, show that the proposed algorithm does not outperform Gibbs sampling in terms of perplexity. Therefore, the question of the importance and applicability of the proposed algorithm in practice remains unclear.

Confidence in this Review

2-Confident (read it all; understood it all reasonably well)


Reviewer 6

Summary

Recently researchers have tried to rethink topic modeling by playing closely with its underlying geometrical structures. Whereas papers along with Arora et al and Anandkumar et al proposed matrix/tensor factorization methods based on spectral inferences, this paper derives a bound (not variational but similarly from relaxation) for LDA's log-likelihood of documents in terms of geometric loss, maximizing the log-likelihood indirectly via minimizing the proposed loss function. The authors argue that this minimization can be achieved by finding the convex hull closest to the set of normalized word count vectors for observed documents. For approximate inference, the GDM algorithm performs weighted k-means clustering first on normalized word count vectors for all documents. Then it finds the intersections between rays (coming from the center of polytope toward each centroid) and the boundaries of surrounding sphere or the target polytope. The tGDM algorithm further finds extreme points by tuning extension parameters and thresholds based on the underlying geometry. The experimental results show that tGDM performs the best being comparable to the state-of-the-art Gibbs sampler in terms of perplexity on held-out documents.

Qualitative Assessment

This paper provides interesting approach for LDA-based topic modeling by explicitly playing with the underlying geometry like other series of recent papers from Arora et al and Anandkumar et al. The main difference is that the authors do not construct higher-order moment matrix, but rather derive a bound for log-likelihood objective in terms of geometric loss. Then instead of finding anchor-words either based on multiple linear programming or greedy methods based on QR decompositions (subset sum selections), they try to find intuitively separated points around the simplex and further improve them in geometrically (tGDM) and nonparametrically (nGDM). As the authors explicitly utilize k-means, the non-parametric version can be simply based on another recent progress DP-means (or MAD-Bayes) rather than stacking up complex multiple bayesian layers. However, there are several concerns which could potentially degrade the paper's arguments and need further verifications. (1) The main log-likelihood function is pure log-likelihood of documents similar to traditional pLSI model rather than including the likelihood terms from Dirichlet parameters, which are Bayesian layers essential to smoothen the model and require harder inference due to the coupling between hidden variables. While several relevant work also treat entries for word-topic matrix simply as parameters, it is crucial to assume document-topic proportions are all statistically sampled from a Dirichlet. In some sense, what the paper proposed could be seen as a new approach to solve pLSI model. (2) There are many reasons why recent work focused on higher-order moment matrix rather than purely decomposing word-document matrix. As proven in Arora et al, the plain word-document matrix is highly susceptible to the noises, whereas 2nd-order word-word co-occurrence matrix is robust, being easily close to the posterior co-occurrence. That is why many researchers try to find the convex hull within the co-occurrence space rather than word-document space. In addition, say M is the number of documents, and V is the size of vocabulary. topic modeling is useful for finding hidden themes for collections with a large M. If M exceeds hundreds of thousands, the proposed method will be also suffered by running-time cost, whereas higher-order moment-based methods still play with complexities scaled only with respect to V. (3) The relaxed assumption in Section 3.1 could impact a lot to the actual topic quality, whereas it might not affect much to the perplexity quantitatively. Having constraints only on the sum of all entries is much weaker than having constraints on each matrix. Have you explored and compared the actual topics generated from your model against other methods? (4) The experiments only includes a corpus with very small number of documents and topics. It would be great if you can test your model on various different corpus with different number of topics. Note that Arora et al's method has been revised by other researchers, showing that the anchor-word algorithm does not perform well for small number of topics in real dataset. Also, trying to show various quantitative measures such as coherence, dissimilarity, and specificity will further help readers' understanding about the results. Minor question: Could you explain why it is a benefit to have a weight based on the length of document toward the loss term?

Confidence in this Review

2-Confident (read it all; understood it all reasonably well)